# Self-Paced Encoding with Adaptive Graph Regularization for Multi-view Clustering

## Abstract

Multi-view graph clustering is a powerful technique for learning discriminative node representations by integrating complementary information from diverse views. However, existing methods often suffer from rigid fusion schemes, ignore sample difficulty during training, and struggle to capture both global semantics and local structures through graph-based regularization. To address these issues, we propose SPEAG, a novel framework for Self-Paced Encoding with Adaptive Graph Regularization. SPEAG combines view-specific graph autoencoders with a unified learning objective that incorporates self-paced training, adaptive view fusion, and structure-aware regularization. Specifically, a self-paced neighborhood expansion strategy is introduced, where the $k$-nearest neighbor graph is gradually densified to learn from easy instances first and hard ones later. Meanwhile, each view's embedding is adaptively weighted based on its importance, and a fusion representation is formed for global consistency. To encourage distributional alignment and enhance cluster compactness, SPEAG integrates a Maximum Mean Discrepancy (MMD) loss across views and a self-supervised clustering objective based on soft assignment refinement. Extensive experiments on real-world datasets demonstrate that SPEAG achieves superior clustering accuracy and robustness compared to existing multi-view graph clustering methods.

## 1 Introduction

Multi-view clustering (MVC) Liu et al. (2022); Fang et al. (2023) seeks to partition unlabeled data by jointly exploiting all views, and recent deep MVC advances leverage powerful neural representations. Representative methods include CoMSC Liu et al. (2021) (feature decomposition for robust representation) and DUA-Nets Geng et al. (2021) (uncertainty-aware view weighting), while CMRL Zheng et al. (2023) and SCMRL Zhou et al. (2023) further explore complementarity and semantic consensus via low-rank tensors and attention. However, many approaches emphasize view-specific features while underutilizing instance–instance relations that are crucial for clustering. Moreover, anchor-based methods reduce computation but often distort local structures, and GNN-based models (e.g., MGCN, MVGRL Kang et al. (2020); Yang & Zhu (2024); Jiang et al. (2025)) frequently fuse views heuristically (e.g., averaging) Chen et al. (2025b) and decouple representation learning from clustering; they also depend on static $k$NN or precomputed similarities Chen et al. (2025a) that are non-adaptive during training and sensitive to noise/outliers.

To address these challenges, we propose a novel Self-Paced and Enhanced Adaptive Graph encoding framework, dubbed SPEAG, for unsupervised multi-view graph clustering. SPEAG introduces several key innovations that are carefully integrated into a unified learning framework:

- Self-paced graph encoding with Laplacian regularization: Instead of a fixed k-NN graph, SPEAG updates each view's adjacency via an encoder–decoder; $k$ increases during training for stable warm-up then global structure, while Laplacian terms preserve local geometry.

- Self-weighted fusion with distribution alignment: Instance-level view weights are learned jointly with embeddings; an MMD loss aligns fused and per-view representations, down-weighting unreliable views and mitigating semantic drift across modalities for more robust multi-view consistency.

- Unified self-supervised clustering, end-to-end: A soft-label clustering loss tightens clusters and feeds back to the encoder; fusion, embedding, and clustering are optimized jointly, enabling mutual reinforcement, efficient cross-feedback, and extensibility within a single training pipeline.

## 2 THE PROPOSED METHOD

In this section, we propose a novel multi-view clustering framework via Self-Paced Encoding with Adaptive Graph regularization (SPEAG), whose crucial details are elaborated.

### 2.1 NOTATIONS

Given $V$ views $\{X^{(v)}\}_{v=1}^V$ with $X^{(v)} \in \mathbb{R}^{N \times d_v}$ and $K$ clusters, where $N$ is the number of samples and $d_v$ the dimension of view $v$, we aim to learn a unified embedding $H \in \mathbb{R}^{N \times d_h}$. SPEAG combines view-specific graph autoencoders with a unified objective featuring self-paced training, adaptive view fusion, and structure-aware regularization. For each view we obtain a latent $Z^{(v)} \in \mathbb{R}^{N \times d_z}$; pairwise distances are $D^{(v)}$, similarities $W^{(v)}$, their symmetrized form $A^{(v)}$, and normalized Laplacian $\hat{L}^{(v)}$. We fuse the view latents into a global embedding $H = \sum_{v=1}^V w^{(v)} Z^{(v)}$, where $w^{(v)}$ denotes the adaptive reliability weight of view $v$ with $w^{(v)} \geq 0$ and $\sum_v w^{(v)} = 1$.

### 2.2 WITHIN-VIEW RECONSTRUCTION

**Graph Embedding Autoencoder**  We employ a graph convolutional autoencoder (GCAE) that ingests the feature matrix and a similarity graph per view. For view $v$, we assume a row-stochastic similarity matrix $\widetilde{W}^{(v)}$ has been constructed (see Section 2.2) and symmetrize it as

$$A^{(v)} = \frac{1}{2}\big(\widetilde{W}^{(v)} + (\widetilde{W}^{(v)})^\top\big). \tag{1}$$

The corresponding degree matrix and (unnormalized) Laplacian are $D_{ii}^{(v)} = \sum_j A_{ij}^{(v)}$ and $L^{(v)} = D^{(v)} - A^{(v)}$, respectively. We further compute the normalized Laplacian

$$\hat{L}^{(v)} = I - (D^{(v)})^{-1/2} A^{(v)} (D^{(v)})^{-1/2} \tag{2}$$

to stabilize message passing in the GCAE.

Feeding $X^{(v)}$ and $\hat{L}^{(v)}$ into the encoder yields the latent representation

$$Z^{(v)} = \hat{L}^{(v)} \phi\big(\hat{L}^{(v)} X^{(v)} W_1^{(v)}\big) W_2^{(v)}, \tag{3}$$

where $W_1^{(v)}, W_2^{(v)}$ are layer parameters and $\phi(\cdot)$ is a nonlinearity. We then reconstruct a row-stochastic similarity from latent distances $\hat{D}_{ij}^{(v)} = \|Z_i^{(v)} - Z_j^{(v)}\|_2^2$ using a per-row softmax

$$\bar{W}_{ij}^{(v)} = \frac{\exp(-\hat{D}_{ij}^{(v)})}{\sum_{j'=1}^N \exp(-\hat{D}_{ij'}^{(v)})}. \tag{4}$$

Reconstruction fidelity is measured by the KL divergence

$$L_{\text{rec}}^{(v)} = D_{\text{KL}}\big(\widetilde{W}^{(v)} \| \bar{W}^{(v)}\big) = \frac{1}{N} \sum_{i,j=1}^N \widetilde{W}_{ij}^{(v)} \log \frac{\widetilde{W}_{ij}^{(v)}}{\bar{W}_{ij}^{(v)}}, \tag{5}$$

which encourages $Z^{(v)}$ to encode the view's graph structure.

**Graph Laplacian Regularization**  In our approach, we incorporate not only the graph structural information but also complementary feature information derived directly from the samples. Under the manifold assumption, if two data points are close in the original high-dimensional space Cai et al. (2008); Wen et al. (2018), their corresponding representations in the learned low-dimensional

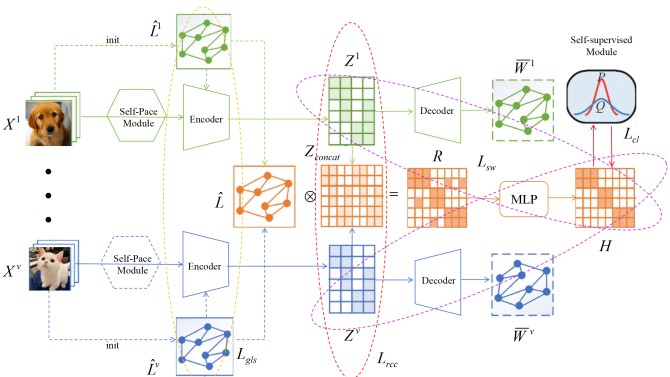

Figure 1: SPEAG adaptively constructs self-paced graphs, encodes with GCN, reconstructs affinities, and fuses multi-view features to enhance clustering via multiple losses.

latent space should also remain close. Concretely, the consensus similarity among different views should be preserved after dimensionality reduction.

To enforce this, we introduce a graph regularization term formulated as follows:

$$L_{\text{lap}}^{(v)} = \sum_{i,j=1}^{N} A_{ij}^v \|z_i^v - z_j^v\|_2^2 = \text{tr}\big((Z^v)^\top L^v Z^v\big), \tag{6}$$

where $A_{ij}^v$ denotes the similarity between samples $i$ and $j$ in the original space of the $v$-th view, $z_i^v$ is the latent representation of sample $i$ in view $v$, and $L^v = D^v - A^v$ is the graph Laplacian matrix for view $v$, with $D^v$ being the corresponding degree matrix. Here, $\text{tr}(\cdot)$ denotes the trace operator, summing the diagonal elements of a matrix. We combine the reconstruction and Laplacian terms as

$$L_{\text{graph}}^{(v)} = L_{\text{rec}}^{(v)} + \lambda_1 L_{\text{lap}}^{(v)}. \tag{7}$$

Intuitively, when $A_{ij}^v$ is large—implying high similarity between samples $i$ and $j$—the regularization penalizes large distances $\|z_i^v - z_j^v\|_2^2$ in the latent space. This encourages similar samples to stay close, preserving local structure and guiding the model to learn embeddings that reflect both feature content and intrinsic neighborhood relationships, thus maintaining the data's manifold structure.

**Self-paced adaptive graph construction**   Inspired by self-paced learning—progressing from easy to hard—we construct the similarity graph progressively. Early training starts from a sparse $k$NN backbone to stabilize optimization, and we gradually enlarge neighborhoods to enrich structure and learn more discriminative representations.

For each view $v$, we maintain a neighborhood size $k_t$ at pre-training epoch $t$, which is increased according to

$$k_t = \min\big(k_0 + t \cdot \Delta k,\ k_{\max}\big), \tag{8}$$

where we use $k_0 = 5$, $\Delta k = 2$, and $k_{\max} = 20$ in all experiments. At $t = 0$ we compute pairwise distances $D_{ij}^{(v)} = \|X_i^{(v)} - X_j^{(v)}\|_2^2$ on the original features; after each pre-training epoch we recompute distances from the current embeddings $Z^{(v)}$ as

$$D_{ij}^{(v)} = \|Z_i^{(v)} - Z_j^{(v)}\|_2^2. \tag{9}$$

For a given epoch $t$, we keep, for each sample $i$, only its $k_t$ nearest neighbors. Non-neighbors have zero similarity, while neighbors use margin-based edge weights

$$\widetilde{W}_{ij}^{(v)} = \frac{D_{i,k_t+1}^{(v)} - D_{ij}^{(v)}}{\sum_{m=1}^{k_t} \big(D_{i,k_t+1}^{(v)} - D_{im}^{(v)}\big)}, \qquad 1 \le j \le k_t, \tag{10}$$

where $D_{i,m}^{(v)}$ denotes the distance between sample $i$ and its $m$-th nearest neighbor. This yields a row-stochastic similarity matrix $\widetilde{W}^{(v)}$ that serves as supervision for the graph autoencoder (Equation 5) and as the basis for the Laplacian regularizer.

The schedule in Equation 8 implements a self-paced densification of the graph: when $k_t$ is small, each node is only connected to its most similar neighbors, which correspond to "easy" and highly confident local relationships. As training proceeds and $k_t$ increases, more distant neighbors with higher uncertainty are gradually incorporated into the graph. In this way, the model first focuses on reliable local structures and then progressively absorbs harder connections, which stabilizes training and improves robustness to noisy views.

## 2.3 MULTI-VIEW FUSION AND CONSISTENCY

**Representational Consistency Constraint**    Given that various perspectives of an object inherently possess consistent characteristics, we enforce this consistency across views through a mechanism referred to as the representational consistency constraint. This constraint promotes alignment among the representations derived from different views, thereby minimizing redundancy and enhancing overall consistency:

$$L_{rcc} = \sum_{v_i, v_j} \left\| Z^{(v_i)} - Z^{(v_j)} \right\|_F^2. \tag{11}$$

This term is only used in the fine-tuning stage and is weighted by $\lambda_4$ in the overall objective.

**Global feature generation**    To integrate complementary information across views and obtain a compact clustering-friendly representation, we aggregate the per-view embeddings $\{Z^{(v)}\}_{v=1}^V$ using learnable reliability weights.

Specifically, we maintain a parameter vector $\mathbf{a} \in \mathbb{R}^V$ and define normalized weights by a softmax

$$w^{(v)} = \frac{\exp(a_v)}{\sum_{u=1}^V \exp(a_u)}, \qquad v = 1, \ldots, V. \tag{12}$$

The fused global embedding is then computed as a weighted sum

$$H = \sum_{v=1}^V w^{(v)} Z^{(v)} \in \mathbb{R}^{N \times d_z}, \tag{13}$$

which directly serves as the consensus representation for clustering.

**Self-weighted Contrastive Learning**    Multi-view contrastive learning has demonstrated strong potential in aligning complementary information from different views. However, conventional methods typically treat all views equally, using uniform weights when computing contrastive losses. Formally, they adopt a view-invariant formulation such as:

$$L_{CL} = \sum_{m,n} L_{CL}^{m,n}(Z^{(m)}, Z^{(n)}), \tag{14}$$

where $Z^{(m)}$ and $Z^{(n)}$ denote the representations of views $m$ and $n$, respectively. While this symmetric formulation facilitates consistency across views, it can undesirably amplify the influence of low-quality or noisy views by forcing them to align equally with high-quality ones. This uniform treatment may lead to representational degeneration and hinder effective feature fusion.

To address this limitation, we propose an inter-view self-weighted contrastive learning strategy that adaptively modulates the contribution of each view based on its semantic alignment with a shared global representation. The core idea is to prioritize reliable, informative views in the contrastive process while suppressing the impact of unreliable ones. Specifically, we reformulate the contrastive loss as:

$$L_{sw} = \sum_{v=1}^V w^{(v)} L_{sw}^{(v)}(Z^{(v)}, H), \tag{15}$$

where $Z^{(v)}$ denotes the view-specific representation, $H$ is the fused global representation, and $w^{(v)}$ is the adaptive weight reflecting the relative reliability of the $v$-th view.

Since labels are unavailable in unsupervised settings, directly evaluating the quality of a view is challenging. To estimate the semantic relevance of each view, we assess the distributional discrepancy between $Z^{(v)}$ and $H$. A lower discrepancy implies a higher alignment with global semantics and thus a more trustworthy view. This discrepancy is denoted as:

$$\mathcal{D}^{(v)} = \mathcal{D}\big(Z^{(v)}, H\big), \tag{16}$$

where $\mathcal{D}(\cdot, \cdot)$ is a distance metric based on Maximum Mean Discrepancy (MMD) Wu et al. (2024), a non-parametric criterion that measures the distance between two distributions in a reproducing kernel Hilbert space (RKHS). Given two feature sets $X_s = \{x_i^s\}_{i=1}^{n_s}$ and $Y_t = \{y_j^t\}_{j=1}^{n_t}$, the squared MMD is defined as:

$$\text{MMD}^2(X_s, Y_t) = \frac{1}{n_s^2} \sum_{i,j=1}^{n_s} k(x_i^s, x_j^s) + \frac{1}{n_t^2} \sum_{i,j=1}^{n_t} k(y_i^t, y_j^t) - \frac{2}{n_s n_t} \sum_{i=1}^{n_s} \sum_{j=1}^{n_t} k(x_i^s, y_j^t), \tag{17}$$

where $k(\cdot, \cdot)$ is a kernel function. In our case, we employ a linear kernel $k(x, y) = x^\top y$, which avoids the need for hyperparameter tuning and suits high-dimensional representations. Given that $Z^{(v)}$ and $H$ share the same dimensions, the discrepancy for each view is computed as:

$$\text{MMD}^2(Z^{(v)}, H) = \frac{1}{N^2} \sum_{i,j=1}^{N} k(Z_i^{(v)}, Z_j^{(v)}) + \frac{1}{N^2} \sum_{i,j=1}^{N} k(H_i, H_j) - \frac{2}{N^2} \sum_{i,j=1}^{N} k(Z_i^{(v)}, H_j), \tag{18}$$

where $N$ denotes the total number of samples. Based on these discrepancies, we define a normalized weight allocation function to adaptively determine the importance of each view:

$$w^{(v)} = \mathcal{P}\big(\mathcal{D}^{(v)}\big) = \text{softmax}\big(-\mathcal{D}^{(v)}\big). \tag{19}$$

The use of the negative discrepancy ensures that views more consistent with global semantics receive higher weights. This adaptive weighting mechanism promotes semantically aligned views and effectively suppresses noisy or misleading ones, thereby enhancing the robustness and expressiveness of the learned global representations.

## 2.4 SELF-SUPERVISED CLUSTERING MODULE

In unsupervised learning, we refine the unified representation $H$ by integrating multi-view information that captures shared and complementary patterns. Since $H$ may not be immediately clustering-friendly, we further enhance it with a self-supervised clustering objective.

**Clustering Loss via KL Divergence**  We adopt a Kullback–Leibler divergence between a target distribution $P$ and a soft assignment $Q$:

$$L_{cl} = D_{\text{KL}}(P\|Q) = \sum_i \sum_j p_{ij} \log \frac{p_{ij}}{q_{ij}}. \tag{20}$$

Here, $Q$ is the soft label distribution and $P$ is the sharpened target; the KL term measures information loss when approximating $P$ by $Q$.

**Soft Label Distribution** $Q$  We compute $q_{ij}$ via a Student-$t$ kernel between feature $h_i$ and centroid $\mu_j$:

$$q_{ij} = \frac{\big(1 + \|h_i - \mu_j\|^2/\sigma^2\big)^{-(\alpha+1)/2}}{\sum_f \big(1 + \|h_i - \mu_f\|^2/\sigma^2\big)^{-(\alpha+1)/2}}, \tag{21}$$

where $\sigma$ controls the kernel scale.

**Target Distribution** $P$  To emphasize confident assignments and balance clusters, we set

$$p_{ij} = \frac{q_{ij}^2/f_j}{\sum_f q_{if}^2/f_f}, \qquad f_j = \sum_i q_{ij}, \tag{22}$$

Table 1: Datasets Descriptions

| Dataset | Clusters | Samples | Dimensionality |
|---|---|---|---|
| COIL20 | 20 | 1140 | [1024, 3304, 6750] |
| Handwritten | 10 | 2000 | [240, 76, 216, 47, 64, 6] |
| HW1256 | 10 | 2000 | [76, 216, 47, 6] |
| Caltech | 7 | 1400 | [40, 254, 1984, 512, 928] |
| MNIST-USPS | 10 | 5000 | [784, 256] |
| Fashion | 10 | 10000 | [784, 784, 784] |

so that larger $q_{ij}$ contributes more while normalizing by cluster frequency.

The final label for node $v_i$ is

$$s_i = \arg\max_j q_{ij}. \tag{23}$$

This self-supervised head aligns $H$ with clustering by sharpening confident assignments, mitigating unreliable signals, and improving separability without external labels.

## 2.5 TRAINING

The training procedure is divided into two main phases: pre-training and subsequent fine-tuning.

**Pre-training stage.** In the pre-training phase, we start from a small neighborhood size $k_0$ and increase it according to Equation equation 8 after each epoch, rebuilding the self-paced $k_t$-NN graphs on the current embeddings. During this stage we only optimize the within-view reconstruction loss and the Laplacian regularizer. Denoting

$$L_{rc} = \sum_{v=1}^{V} L_{rec}^{(v)}, \qquad L_{gls} = \sum_{v=1}^{V} L_{lap}^{(v)},$$

the preliminary training loss is given by:

$$L_{pre} = L_{rc} + \lambda_1 L_{gls}. \tag{24}$$

**Fine-tuning stage.** In the fine-tuning phase, the self-paced graphs are held fixed, and we enforce inter-view consistency and clustering-friendliness. The model is refined by minimizing the following loss:

$$L_{fine} = L_{rc} + \lambda_1 L_{gls} + \lambda_2 L_{rcc} + \lambda_3 L_{sw} + \lambda_4 L_{cl}. \tag{25}$$

Here, $\lambda_1$, $\lambda_2$, $\lambda_3$ and $\lambda_4$ are coefficients that regulate the impact of the graph-based smoothness ($L_{gls}$), representational consistency ($L_{rcc}$), self-weighted contrastive learning ($L_{sw}$), and self-supervised clustering ($L_{cl}$) terms within the total loss function, respectively. Ultimately, we apply the Self-supervised Clustering Module to the consolidated representation $H$ to derive the clustering outcomes.

## 3 EXPERIMENTS

### 3.1 DATASETS

COIL20 comprises grayscale images of 20 objects across 360° poses. Handwritten and HW1256 are multi-view handwritten digits (differing in the number of views). Caltech contains multi-feature object/scene images. MNIST-USPS mixes two digit sources to form a cross-domain benchmark. Fashion consists of clothing images with multiple attributes/views. Cluster counts, sample sizes, and view dimensionalities are in Table1.

Table 2: Clustering Results on COIL20, Handwritten, HW1256 and MNIST-USPS Datasets

| Dataset | COIL20 | | Handwritten | | HW1256 | | Caltech | | MNIST-USPS | | Fashion | |
|---|---|---|---|---|---|---|---|---|---|---|---|---|
| | ACC | NMI | ACC | NMI | ACC | NMI | ACC | NMI | ACC | NMI | ACC | NMI |
| DUA-Nets | 0.7228 | 0.8272 | 0.6585 | 0.5924 | 0.7425 | 0.7933 | 0.5461 | 0.0154 | 0.9136 | 0.8359 | 0.7747 | 0.8145 |
| SGFCC | 0.2590 | 0.4381 | 0.3870 | 0.5501 | 0.3840 | 0.5118 | 0.4817 | 0.5262 | 0.9526 | 0.9412 | 0.9286 | 0.9180 |
| CoMSC | 0.5482 | 0.7382 | 0.5881 | 0.4914 | 0.7320 | 0.6793 | 0.4105 | 0.4830 | 0.7252 | 0.7025 | 0.6050 | 0.7158 |
| CMRL | 0.6264 | 0.7575 | 0.5439 | 0.4865 | 0.8947 | 0.8168 | 0.4082 | 0.3399 | 0.9308 | 0.8690 | 0.5483 | 0.6134 |
| ASR-ETR | 0.6611 | 0.7940 | 0.7580 | 0.6930 | 0.7290 | 0.6487 | 0.5096 | 0.5133 | 0.7580 | 0.6930 | 0.7186 | 0.7351 |
| RCAGL | 0.6701 | 0.8127 | 0.8775 | 0.8061 | 0.9305 | 0.8623 | 0.6341 | 0.4871 | 0.8925 | 0.7316 | 0.7924 | 0.8097 |
| HFMVC | 0.4558 | 0.5956 | 0.9080 | 0.8341 | 0.8785 | 0.7927 | 0.5863 | 0.3280 | 0.9010 | 0.8431 | 0.9110 | 0.9008 |
| GCFAgg | 0.3458 | 0.4886 | 0.8085 | 0.7752 | 0.8005 | 0.7664 | 0.3813 | 0.4321 | 0.9300 | 0.8896 | 0.8982 | 0.8714 |
| SCMVC | 0.5153 | 0.6451 | 0.8945 | 0.8168 | 0.7945 | 0.7047 | 0.4905 | 0.4390 | 0.9576 | 0.9505 | 0.9229 | **0.9213** |
| DCMVC | 0.7340 | 0.8162 | 0.8995 | 0.8718 | 0.7580 | 0.7620 | 0.3161 | 0.2460 | 0.8920 | 0.9059 | 0.7836 | 0.8745 |
| DDMVC | 0.9016 | 0.9515 | 0.8840 | 0.7727 | 0.9318 | 0.8938 | 0.5814 | 0.4752 | 0.9324 | 0.9190 | 0.9112 | 0.9032 |
| RTGD-MVC | 0.8765 | 0.9090 | 0.8863 | 0.8100 | 0.9324 | 0.8981 | 0.6551 | 0.4718 | 0.9515 | 0.9422 | 0.9124 | 0.8930 |
| Ours | **0.9153** | **0.9651** | **0.9115** | **0.8467** | **0.9560** | **0.9145** | **0.6679** | **0.5345** | **0.9628** | **0.9515** | **0.9328** | 0.8935 |

## 3.2 COMPARATIVE ALGORITHMS

Baselines fall into three groups: *(i) adaptive weighting/uncertainty* (DUA-Nets Geng et al. (2021), RCAGL Liu et al. (2024), SCMVC Wu et al. (2024)), which modulate view contributions by reliability; *(ii) subspace/anchor representations* (CoMSC Liu et al. (2021), CMRL Zheng et al. (2023), AER-ETR Ji & Feng (2023)) to reduce redundancy via compact bases; and *(iii) contrastive/structural constraints* (HFMVC Jiang et al. (2024), DCMVC Cui et al. (2024), GCFAgg Yan et al. (2023), SGFCC Shu et al. (2024)), DDMVC Xu et al. (2025), RTGD-MVC Zou et al. (2025) to enforce cross-view consistency and cluster structure. Most do not jointly leverage **graph-structural guidance** with **contrastive consistency**; SPEAG unifies both.

## 3.3 COMPLEXITY AND EFFICIENCY

We briefly analyze the computational complexity of SPEAG. Let $N$ be the number of samples, $V$ the number of views, $d$ the embedding dimension and $k_t$ the neighborhood size at epoch $t$.

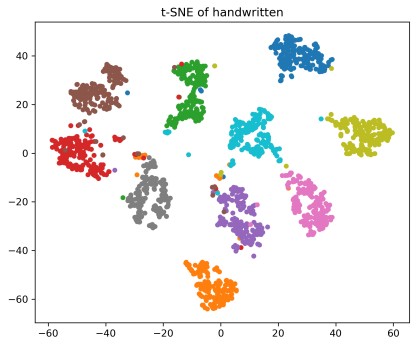
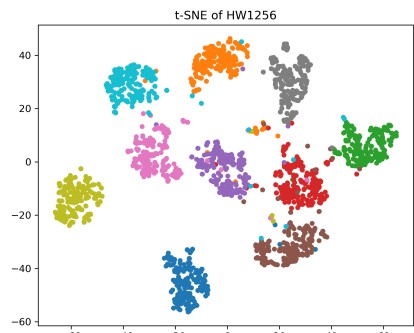

Figure 2: T-SNE visualization on the datasets handwritten and HW125

**Time complexity.** Constructing the $k_t$-NN graph for view $v$ requires computing pairwise distances between $N$ samples and selecting the $k_t$ nearest neighbors for each sample. With a naive implementation, this takes $\mathcal{O}(VN^2d)$ time per graph update, which is further reduced in practice by mini-batch processing and efficient $k$NN routines. Given the self-paced schedule in Eq. equation 8, we update the graph only once per pre-training epoch.

The forward and backward passes of the GCAE layers scale as $\mathcal{O}(VNk_td)$ since each node aggregates messages from at most $k_t$ neighbors. The MMD-based weighting and contrastive loss operate on the embeddings and have time complexity $\mathcal{O}(VN^2)$ in the worst case, but can be implemented in a mini-batch fashion.

Table 3: Training time comparison (in seconds) on representative datasets. 'Pre' and 'Fine' denote the pre-training and fine-tuning stages, respectively.

| Dataset | Method | Pre ep. | Pre time/ep. | Fine ep. | Fine time/ep. | Total time |
|---------|--------|---------|--------------|----------|---------------|------------|
| HW1256 | DDMVC | 100 | 1.444 | 200 | 2.381 | 620.689 |
| | RTGD-MVC | 100 | 1.398 | 200 | 2.378 | 615.453 |
| | **SPEAG** | 200 | 2.365 | 200 | 3.454 | 927.186 |
| MNIST-USPS | DDMVC | 100 | 1.895 | 200 | 3.064 | 802.328 |
| | RTGD-MVC | 100 | 2.368 | 200 | 3.420 | 920.838 |
| | **SPEAG** | 200 | 2.900 | 200 | 4.067 | 1103.372 |

**Memory complexity.** The main memory cost of SPEAG comes from storing the multi-view embeddings $\{Z^{(v)}\}_{v=1}^V$ and the sparse $k_t$-NN graphs. The space complexity is $\mathcal{O}(VNd + VNk_t)$, which is comparable to other graph-based deep clustering methods.

To further quantify the efficiency of SPEAG, we report the wall-clock training time on representative datasets in Table 3.

## 3.4 MODEL ANALYSIS

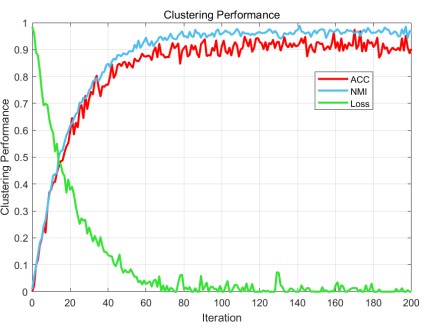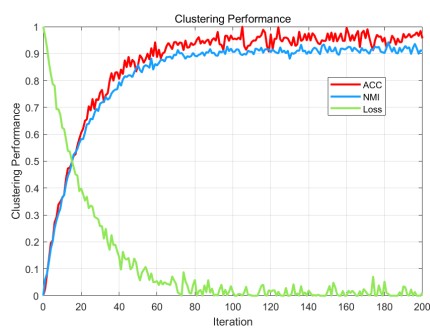

Figure 3: Clustering performance with increasing iteration on COIL20 and HW1256

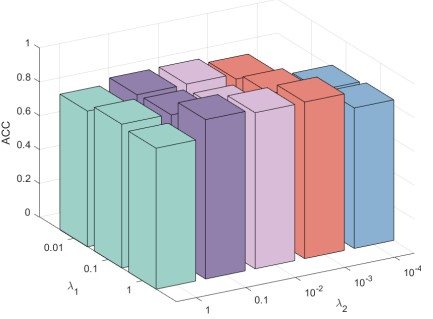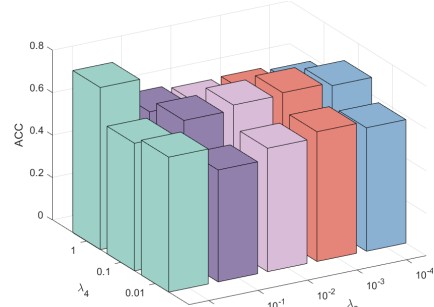

Figure 4: ACC sensitivity on Caltech: left—$\lambda_1, \lambda_2$; right—$\lambda_3, \lambda_4$.

**Performance Evaluation** We evaluate on six benchmarks using ACC/NMI (Table 3). Findings: **(1)** SPEAG achieves best or second-best results on most datasets, driven by self-paced graph construction and structure-aware contrastive learning; **(2)** versus shallow/hybrid methods (KMeans, CoMSC, ASR-ETR, RCAGL), SPEAG better captures nonlinear cross-view relations—particularly strong on image datasets (MNIST-USPS, Fashion); **(3)** compared with deep baselines (DUA-Nets, CMRL, HFMVC, SCMVC, DCMVC, GCFAgg, SGFCC), SPEAG augments contrastive alignment with explicit graph supervision, yielding more clustering-friendly embeddings than methods that emphasize only consistency or only contrast.

Table 4: Ablation study of each loss term on three datasets.

| Loss terms | | | | COIL20 | | | HW1256 | | | MNIST-USPS | | |
|:---:|:---:|:---:|:---:|:---:|:---:|:---:|:---:|:---:|:---:|:---:|:---:|:---:|
| $\mathcal{L}_{gls}$ | $\mathcal{L}_{rcc}$ | $\mathcal{L}_{sw}$ | $\mathcal{L}_{cl}$ | ACC | NMI | ARI | ACC | NMI | ARI | ACC | NMI | ARI |
| ✓ | ✓ | ✓ | | 81.01 | 87.63 | 77.38 | 83.85 | 78.37 | 71.49 | – | – | – |
| ✓ | ✓ | | ✓ | 87.57 | 94.55 | 86.23 | 77.35 | 74.86 | 67.72 | 68.14 | 62.92 | 53.34 |
| ✓ | | ✓ | ✓ | 81.04 | 87.32 | 75.86 | 76.85 | 73.91 | 66.87 | 62.82 | 55.50 | 43.49 |
| | ✓ | ✓ | ✓ | 81.04 | 87.32 | 75.86 | 76.75 | 74.51 | 67.36 | 68.44 | 63.09 | 53.39 |
| ✓ | ✓ | ✓ | ✓ | **91.53** | **96.51** | **90.87** | **95.60** | **91.45** | **90.37** | **96.28** | **95.15** | **90.39** |

Table 5: Impact of fixed $k$ (number of neighbor) and increased $k$ on clustering results for four datasets. For each dataset, we select eight values equidistantly between 0 and $k_{\max}$ as fixed $k$. We report the ACC metric.

| | COIL20 | | Handwritten | | HW1256 | | MNIST-USPS |
|:---:|:---:|:---:|:---:|:---:|:---:|:---:|:---:|
| $k$ | ACC | $k$ | ACC | $k$ | ACC | $k$ | ACC |
| 2 | $69.30 \pm 1.01$ | 5 | $\mathbf{91.15 \pm 1.15}$ | 5 | $52.58 \pm 4.96$ | 10 | $89.38 \pm 1.03$ |
| 4 | $84.84 \pm 1.19$ | 10 | $88.83 \pm 1.27$ | 10 | $79.72 \pm 3.12$ | 20 | $92.18 \pm 1.15$ |
| 6 | $87.11 \pm 1.29$ | 15 | $87.05 \pm 0.13$ | 15 | $78.27 \pm 4.04$ | 30 | $93.63 \pm 0.87$ |
| 8 | $89.82 \pm 1.37$ | 20 | $81.17 \pm 1.04$ | 20 | $76.57 \pm 4.13$ | 40 | $93.91 \pm 0.79$ |
| 10 | $90.84 \pm 1.11$ | 25 | $78.39 \pm 0.86$ | 25 | $91.51 \pm 0.63$ | 50 | $95.24 \pm 0.68$ |
| 12 | $\mathbf{91.53 \pm 1.09}$ | 30 | $78.28 \pm 0.66$ | 30 | $\mathbf{95.60 \pm 4.21}$ | 60 | $\mathbf{96.28 \pm 0.55}$ |
| 14 | $88.70 \pm 0.73$ | 35 | $77.72 \pm 0.82$ | 35 | $88.59 \pm 1.63$ | 70 | $94.30 \pm 0.48$ |
| 16 | $86.36 \pm 1.06$ | 40 | $77.23 \pm 1.10$ | 40 | $92.57 \pm 2.54$ | 80 | $94.12 \pm 0.42$ |

**Ablation Study**  We study four losses on COIL20: graph regularization $\mathcal{L}_{gls}$, cross-view consistency $\mathcal{L}_{rcc}$, self-weighted contrastive $\mathcal{L}_{sw}$, and self-supervised clustering $\mathcal{L}_{cl}$. Results show $\mathcal{L}_{gls}$ notably improves clustering; removing any fine-tuning loss degrades performance—most severely without $\mathcal{L}_{cl}$ (weaker instance discrimination). Dropping $\mathcal{L}_{sw}$ harms cross-view distribution alignment, and dropping $\mathcal{L}_{rcc}$ weakens structural consistency. The full SPEAG model is best.

**Parameters and Convergence Analysis**  As iterations increase (Fig. 3), ACC/NMI rise and the loss decreases, indicating stable convergence and continuous improvement. Fig. 2 shows hyperparameter sensitivity: $\lambda_1$ and $\lambda_3$ have stronger effects; within reasonable ranges, larger values generally yield more robust gains.

## 4 CONCLUSION

In this work, we have presented SPEAG, a novel self-paced exemplar-aware graph learning framework for multi-view clustering. By integrating an exemplar-guided attention mechanism with a self-paced training strategy, SPEAG effectively balances the exploration of consistent and complementary information across views while progressively mitigating the impact of noisy or low-quality samples. Moreover, the joint learning of view-specific and consensus representations within a unified anchor graph structure allows for more robust clustering performance. Extensive experiments on multiple benchmark datasets demonstrate that our method achieves competitive or superior results compared to state-of-the-art approaches. In future work, we plan to extend SPEAG to handle streaming or dynamically evolving multi-view data, and explore its potential in semi-supervised and federated clustering scenarios.

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
