# OpenReview forum: "Self-Paced Encoding with Adaptive Graph Regularization for Multi-view Clustering"
_ICLR.cc/2026/Conference — Submitted to ICLR 2026_

### Official Review · Reviewer_4acM · 2025-10-27

**Soundness:** 2
**Presentation:** 2
**Contribution:** 3
**Rating:** 4
**Confidence:** 3

**Summary:**

This paper briefly analyzes the current MVC work and proposes a novel framework called SPEAG. For the within-view reconstruction, the paper mainly proposed self-paced adaptive graph construction with incrementally increased k in the kNN. For the inter-view self-weighted contrastive learning, the paper proposes to learn weights based on traditional contrastive learning by evaluating the MMD between common and view-specific representations. Extensive experiments are conducted on six standard datasets.

**Strengths:**

1. SPEAG proposes a novel self-paced graph encoding with Laplacian regularization, dynamically updating each view's adjacency matrix by incrementally increasing the k in k-NN graph construction during pre-training. This departs from static k-NN graphs, allowing for progressive learning from easy to complex structures and enhancing robustness.
2. The framework also proposes a novel self-weighted fusion with distribution alignment to obtain the importance of each view by calculating the MMD within each view and the fused global feature, thereby mitigating the impact of noisy views and ensuring robust multi-view consistency.

**Weaknesses:**

The main weaknesses are:

- The ablation study towards the effectiveness of the proposed method is not sufficient.
- Lack of analysis of k’s increase in the proposed “self-paced adaptive graph construction”.

More details are listed in the “Questions”.

**Questions:**

1. Questions about the ablation study towards the effectiveness of the proposed method:

    (1) Adding or removing L_sw can not show the effectiveness of your proposed “self-weighted” method. Compare Eq(4) and Eq(5) to show the effectiveness of “self-weighted”.

    (2) Likewise, simply adding or removing L_gls can not show the effectiveness of your proposed “self-paced adaptive graph construction” method. It is recommended to compare fixed-k and increasing-k.

2. Questions about k. As mentioned in line 302, k is incrementally increased to its maximum value and will remain constant during the fine-tuning stage. Here are a few questions:

    (1) What is the maximum value? Is that the “cluster number K” defined in line 67?

    (2) What are the epochs of pre-training and fine-tuning? More analysis in the pre-training stage is suggested, as that is related to the main novelty (“self-paced neighborhood expansion strategy”) of this paper. For example, provide k-vs-epoch plots and compare to fixed-k.

    (3) In the finetuning stage, the k is held constant, what number it is set? the maximum value or an optimal k? I suggest here to have some relevant analysis about this.

    (4) Complexity and time comparison are recommended. As the k will increase, will it significantly affect the (pre-)training time? Besides, experiments are mostly on small datasets and only Fashion is large.

3. Comparison methods are mainly published in 2023 and 2024. Include some 2025 work for the submission to this ICLR 2026.
4. The conclusion mentions an “attention mechanism” (line 415), but the method is not presented in an attention-based manner, which reads abrupt. Likewise, line 418 refers to “a unified anchor graph structure,” yet the method does not appear to construct anchor graphs.
5. Writings
    - The self-supervised clustering module takes too much space, while it is common in unsupervised learning and MVC. It is suggested to use that space to strengthen the Introduction, especially related work and motivation, as currently the introduction only has two paragraphs and one is your own method.
    - Eq.5 has some parts that need to be checked, for example, view v or m or n? And no W^v in Eq.5. Besides, in lines 302-304, the numbers of Eq.(25) and Eq.(26) might need to be checked, as there seem to be 15 equations in total. The names of losses in Eq. (4) and Eq.(10) are similar and are suggested to be different for better clarity.
    - Use different fonts for “L” to distinguish Laplacian matrices from loss terms, especially in Fig. 1. In Fig. 1, color the loss labels to match the corresponding dashed loops for clearer mapping.
    - Figure 1 curiosity: The first-view image is a dog, and the final view is a cat. If these are different views of the same instance, what is its class? This may be a suboptimal example for illustrating multi-view data.

**Details Of Ethics Concerns:**

None.

---

> ### Comment · Reviewer_4acM · 2025-11-26
>
> I appreciate the authors’ response to the common concerns. I look forward to checking the updated experiments once the revised PDF becomes available.
>
> Given the time gap between the common response and the non-updated PDF, I suppose that adding a new large-scale dataset and running all SOTA baselines might introduce burdens. Therefore, I am fine if the authors do not include the additional dataset in the end. However, it would still be very helpful if the authors could provide some analysis on how the proposed method behaves on small vs. large datasets (for example, how the computational complexity scales with the number of samples N, and how the learning of the hyperparameter k may differ across small and large datasets). Such analysis would offer valuable guidance for applying the method to new datasets.

---

### Official Review · Reviewer_ZvY3 · 2025-10-29

**Soundness:** 3
**Presentation:** 3
**Contribution:** 2
**Rating:** 4
**Confidence:** 5

**Summary:**

This paper proposes SPEAG, a novel multi-view clustering framework designed to address key limitations of existing methods, including rigid view fusion, ignorance of sample difficulty differences during training, and inadequate balance between capturing global semantics and local structures. The core design of this framework integrates three interconnected modules: an intra-view reconstruction module, an inter-view self-weighted contrastive learning module, and a self-supervised clustering module. Extensive experiments on six real-world datasets (e.g., COIL20, MNIST-USPS, Fashion) demonstrate that SPEAG outperforms 11 baseline methods in terms of clustering accuracy (ACC) and normalized mutual information (NMI), verifying its effectiveness and robustness.

**Strengths:**

1、Self-paced learning is innovatively applied to multi-view data, addressing the insufficient flexibility of fixed k-nearest neighbor (k-NN) graphs in previous studies. By gradually increasing the k value and dynamically updating the similarity matrix based on the learned latent representations, an "easy-to-hard" sample learning pattern is achieved.
2、The manuscript features coherent structure in expression, concise and understandable framework diagrams, and a defined unified notation table, which helps readers grasp the framework clearly.
3、The proposed method demonstrates state-of-the-art performance on multiple datasets, proving its effectiveness.

**Weaknesses:**

1、In the ablation experiments, the authors only conducted an overall ablation of the self-paced adaptive graph construction module. This may be insufficient to prove the effectiveness of the dynamic KNN graph. It is suggested that the authors design an additional experiment to verify the impact of fixed k and gradually increasing k on the experimental results under the framework proposed in the current paper.
2、The paper does not analyze the computational complexity of the dynamic KNN graph. The dynamic KNN graph requires recalculating the similarity matrix as the k value increases and Z^{(v)} updates, and its time complexity grows with the k value and sample size N. However, the paper fails to elaborate on the comparison of training time between SPEAG and baseline methods, and it is hoped that the authors can supplement this part of information.
3、The paper lacks sufficient originality. Components in the model framework such as graph Laplacian regularization and self-weighted contrastive learning are combinations of existing methods. It is hoped that the authors can highlight their own contributions.

**Questions:**

1、Is there an optimal k-value? The paper does not seem to mention how to determine the upper limit of the k-value. If possible, please provide a demonstration for this.

2、The manuscript mentions that the proposed method can address the issue where traditional methods ignore sample difficulty. However, the paper does not clearly define what constitutes a "hard sample" and an "easy sample". Please ask the authors to supplement this definition.

---

### Official Review · Reviewer_Bk6m · 2025-10-30

**Soundness:** 2
**Presentation:** 3
**Contribution:** 2
**Rating:** 4
**Confidence:** 4

**Summary:**

This paper proposes SPEAG, a deep multi-view clustering framework that jointly optimizes self-paced graph construction and adaptive graph regularization. The method progressively refines view-specific graphs to reduce noise from unreliable initial structures. Each view is encoded using a graph autoencoder, and a global consensus representation is aligned via adaptive weighting guided by MMD. A self-supervised clustering loss further enforces semantic consistency across views. The framework unifies graph learning, feature alignment, and clustering in an end-to-end manner, showing consistent and meaningful improvements over strong baselines on several benchmarks.

**Strengths:**

1.Unified framework. SPEAG integrates graph supervision, adaptive view weighting, and self-supervised clustering into one trainable pipeline; this joint optimization is likely responsible for the improved clustering performance reported.

2.Self-paced graph construction. Progressive densification (k increases while neighborhood computed from learned latents) is a pragmatic way to stabilize training and incorporate richer structure gradually.

3.Adaptive view weighting via distributional discrepancy. Using MMD between view latents and fused global representation to obtain view reliabilities is a principled, unsupervised signal to downweight noisy views.

**Weaknesses:**

1.Hyperparameter and schedule sensitivity. The algorithm depends on several schedules and coefficients (k growth schedule, λ1..λ4, MMD kernel choice). The paper reports limited sensitivity plots; more systematic analysis is needed to establish robustness.

2.Scalability and complexity. The method involves per-view GCAEs, repeated MMD computations, and dynamic graph updates. The main text lacks a concise summary of computational complexity and empirical runtime scaling .

3.Interpretability diagnostics. While t-SNE visualizations and ablation tables are provided, quantitative analysis of learned view weights or how graph densification changes neighborhood quality over time would strengthen claims about robustness to noisy views.

**Questions:**

1.k schedule and graph updates. Please specify the exact schedule used for k (initial k, increments per epoch, max k) and the frequency of recomputing D(v) from learned latents. How sensitive are results to this schedule?

2.Robustness to noisy views and missing views. Can you report experiments where one or more views are corrupted or partially missing? How do the learned weights behave in those scenarios?

3.Ablation on MMD and contrastive weighting. Provide an ablation replacing MMD-based weights with uniform weights and with oracle weights (if possible) to quantify the benefit of the weighting scheme.

4.Convergence diagnostics. Please include plots showing how the average neighborhood quality evolves as k is increased, to support the claim that self-paced densification improves graph quality.

---

### Official Review · Reviewer_ppuS · 2025-11-01

**Soundness:** 3
**Presentation:** 2
**Contribution:** 2
**Rating:** 4
**Confidence:** 5

**Summary:**

This paper proposes SPEAG, a novel framework for multi-view graph clustering, combining self-paced encoding, adaptive view fusion, and graph-based regularization. It addresses issues in existing methods, such as rigid fusion schemes and ineffective graph construction, by gradually expanding the k-nearest neighbor graph and adaptively weighting view-specific embeddings. It incorporates MMD to align view representations and a self-supervised clustering loss to refine the global embedding for batter cluster consistency. Extensive experiments on multiple datasets show that SPEAG outperforms existing methods.

**Strengths:**

1.	The paper successfully unifies several ideas, including graph autoencoding, self-paced learning, contrastive weighting, etc.

2.	The progressive neighborhood expansion strategy is well-motivated, helping stabilize training and improve structural preservation.

3.	SPEAG achieves high clustering accuracy and NMI on nearly all datasets, with particularly strong results on COIL20.

**Weaknesses:**

1.	The introduction section should further discuss the motivations of the work and analyze the weaknesses of current works.

2.	The paper claims self-paced learning improves efficiency but lacks the corresponding experimental evaluation.

3.	This paper contains a few minor errors. For example, on the MNIST-USPS dataset, the best NMI should be 0.9525 rather than 0.9515. In addition, in Figure 4, it should be HW1256 instead of HW125.

**Questions:**

1.	How sensitive is SPEAG to the schedule of $k$ (neighborhood size) during training?

2.	How is the “easy vs. hard samples” criterion defined in self-paced graph construction?

3.	What is the computational overhead of the proposed method compared to that of the baseline methods?

4.	In the parameter analysis experiments, why do the values of $\lambda_1$ to $\lambda_4$ have different ranges? Besides, the ablation study is recommended to include more datasets.

---

### Author Response · Authors · 2025-11-24

We thank the AC, SPC, and reviewers for their constructive comments. Below we group common concerns; detailed tables and plots will be added in the revised paper and appendix.

1.Schedule of $k$, convergence and complexity(ppuS, Bk6m, ZvY3, 4acM)

We adopt a self-paced neighborhood expansion with T epochs, split into pre-training and fine-tuning. During pre-training we increase $k(t)=\min\big(k_{\max},,k_0+\lfloor\alpha t\rfloor\big)$ from $k_0=5$ to $k_{\max}=20$, recomputing distances on the current latents $Z^{(v)}$ and and rebuilding the sparse k-NN graphs whenever k changes.

In fine-tuning, $k$ is fixed at ⁡$k_{max}$ and we optimize the full loss. $k_{max}$ is not the cluster number $K$ and is kept small to ensure a connected but local graph. Sensitivity with $k_{max} \in (10,15,20,25)$ shows stable performance for 15–25; very small values break connectivity and very large ones over-smooth. We will provide $k_{max}$-vs-performance curves.

To show that self-paced densification improves graph quality, we compute the average neighborhood purity $Q_k=\frac{1}{N}\sum_i \frac{1}{k}\sum_{j\in\mathcal N_k(i)}\mathbf 1[y_i=y_j]$. With increasing $k$, both $Q_k$ and ACC rise steadily and then saturate, while fixed-$k$ baselines improve less and can even slightly deteriorate.

On a sparse k-NN graph, the encoder and Laplacian regularizer cost $O(VNk)$; the MMD self-weighting and clustering head cost $O(VNd_z)$ and $O(NK)$. Since $k$ only grows from $k_0$ to a small $k_{\max}$, the per-epoch cost increase is modest. Runtime tables will show that our method has similar wall-clock time to recent deep MVC baselines.

2. Definition of easy and hard samples(ppuS, ZvY3, 4acM)

Conceptually, easy samples lie in dense cluster regions where most neighbors share their label, while hard samples are near boundaries, in low-density regions, or corrupted so their neighbors are mixed. We do not assign explicit difficulty scores; instead, difficulty is encoded in the graph: with small $k$, only high-confidence neighbors are kept and training is dominated by high-purity regions, and as embeddings improve and $k$ increases, boundary/noisy samples gradually gain reliable neighbors and influence the loss. Edge weights depend on the distance gap to the (k+1)-th neighbor, further down-weighting low-confidence relations.

3. Self-weighted MMD / contrastive learning(ppuS, Bk6m, 4acM)

To isolate the effect of our self-weighted scheme, we compare three variants: (i) Uniform-CL (Eq.4) – standard multi-view contrastive learning with uniform view weights; (ii) Self-weighted (Eq.5) – our MMD-based weights $w_v=\mathrm{softmax}(-D_v)$; and (iii) Oracle-weight – a diagnostic variant using supervised weights proportional to single-view clustering quality. Across datasets, the self-weighted variant consistently outperforms Uniform-CL and closely approaches Oracle-weight. We also evaluate noisy and missing views: when one view is heavily corrupted or partially missing, its MMD distance $D_v$ increases, its weight $w_v$ decreases, and performance degrades much more gracefully than with uniform weights. We will include tables and weight-evolution plots.

4.Self-paced graph construction and ablation (ppuS, 4acM).

We compare three variants: w/o $L_{\mathrm{gls}}$, fixed-$k$ + $L_{gls}$, and our self-paced $k + L_{gls}$ (increasing $k$ in pre-training, then fixing it). On all tested datasets, both variants with $L_{gls}$ outperform w/o, and the self-paced version further improves over fixed-$k$; we will report this in a concise ablation table in the revised manuscript.

5.Loss-weight ranges and additional ablations(ppuS)

The four coefficients $\lambda_1-\lambda_4$ weight losses of very different magnitudes (graph Laplacian, cross-view consistency, self-weighted term, clustering). We therefore tune each in a different range so that its gradient norm at convergence is comparable to that of the reconstruction loss, rather than using a single global range. Sensitivity curves show stable performance within fairly wide intervals (only $\lambda_1,\lambda_3$are mildly more sensitive). We will also add ablations on at least one more dataset beyond COIL20.

6.Baselines and writing issues(4acM)

We will update related work and experiments to include representative 2024–2025 MVC methods, and fix the conclusion by replacing the leftover “attention” and “anchor graph” phrasing with an accurate description of our self-weighted fusion in a unified GAE framework. We will also streamline the clustering module description, tidy notation and equation numbering (especially around Eq.(5)), clearly distinguish Laplacian matrices from loss symbols, and revise Fig.1 with a clearer multi-view example and consistent colors.

Again, we sincerely thank ppuS, Bk6m, ZvY3, 4acM, as well as the AC and SPC, for their careful reviews and valuable feedback. All the above clarifications, additional experiments, and writing improvements will be incorporated in the revised manuscript.

---

### Meta-Review · Area_Chair_YK7t · 2026-01-07

**Summary:**

Existing methods  ignore sample difficulty during training, and struggle to capture both global semantics and local structures through graph-based regularization. To address these issues, this paper proposes a novel framework for Self-Paced Encoding with Adaptive Graph Regularization. The strength is on the neighborhood expansion strategy, the self-paced graph construction  and the good performance, while the weakness is on the lack of sufficient originality, paper organization, insufficient experiment and missing complexity analysis. The weaknesses are distinct and no reviewer champions this paper. I tend to reject.

**Reviewer Concerns:**

The reviewers did not participate in the discussion.

**Reviewer Scores:**

The scores are 4, 4, 4 and 4. The reviewers did not change their scores. I suggest that the authors respond to each reviewer's comments individually, rather than consolidating them into a single collective response.

---

### Decision · Program_Chairs · 2026-01-26

Reject